# Effects on maternal and pregnancy outcomes of first-trimester malaria infection among nulliparous women from Kenya, Zambia, and the Democratic Republic of the Congo

**Sequoia I. Leuba** [1]*, **Daniel Westreich** [1], **Carl L. Bose**[2], **Andrew F. Olshan**[1], **Steve M. Taylor**[3], **Antoinette Tshefu**[4], **Adrien Lokangaka**[4], **Waldemar A. Carlo**[5], **Elwyn Chomba**[6], **Musaku Mwenechanya**[6], **Edward A. Liechty**[7], **Sherri L. Bucher** [7], **Osayame A. Ekhaguere**[7], **Fabian Esamai**[8], **Paul Nyongesa**[8], **Saleem Jessani**[9], **Sarah Saleem**[9], **Robert L. Goldenberg**[10], **Janet L. Moore**[11], **Tracy L. Nolen**[11], **Jennifer Hemingway-Foday**[11], **Elizabeth M. McClure**[11], **Marion Koso-Thomas**[12], **Richard J. Derman**[13], **Matthew Hoffman**[14], **Steven R. Meshnick**[1†], **Melissa Bauserman**[2]*

**1** Department of Epidemiology, University of North Carolina-Chapel Hill, Chapel Hill, North Carolina, United States of America, **2** Department of Pediatrics, University of North Carolina -Chapel Hill, Chapel Hill, North Carolina United States of America, **3** Division of Infectious Diseases and Duke Global Health Institute, Duke University Medical Center, Durham, North Carolina, United States of America, **4** Kinshasa School of Public Health, Kinshasa, Democratic Republic of the Congo, **5** University of Alabama at Birmingham, Birmingham, Alabama, United States of America, **6** University Teaching Hospital, Lusaka, Zambia, **7** School of Medicine, Indiana University, Indianapolis, Indiana, United States of America, **8** Department of Child Health and Paediatrics, Moi University School of Medicine, Eldoret, Kenya, **9** Department of Community Health Sciences, Aga Khan University, Karachi, Pakistan, **10** Department of Obstetrics and Gynecology, Columbia University, New York, New York, United States of America, **11** Social, Statistical and Environmental Sciences, RTI International, Research Triangle Park, North Carolina, United States of America, **12** *Eunice Kennedy Shriver* National Institute of Child Health and Human Development, Bethesda, Maryland, United States of America, **13** Thomas Jefferson University, Philadelphia, Pennsylvania, United States of America, **14** Department of Obstetrics and Gynecology, Christiana Care, Newark, Delaware, United States of America

† Deceased.

\* Sequoia.leuba@gmail.com (SIL); Melissa_bauserman@med.unc.edu (MB)

## Abstract

### Background

Few studies have assessed the impact of first-trimester malaria infection during pregnancy. We estimated this impact on adverse maternal and pregnancy outcomes.

### Methods

In a convenience sample of women from the ASPIRIN (Aspirin Supplementation for Pregnancy Indicated risk Reduction In Nulliparas) trial in Kenya, Zambia, and the Democratic Republic of the Congo, we tested for first-trimester *Plasmodium falciparum* infection using quantitative polymerase chain reaction. We estimated site-specific effects on pregnancy outcomes using parametric g-computation.

### Results

Compared to uninfected women, we observed the adjusted site-specific prevalence differences (PDs) among women with first-trimester malaria of the following pregnancy

**Data Availability Statement:** De-identified participant data collected for the study are available

at the NICHD Data and Specimen Hub (https://dash.nichd.nih.gov/study/416283).

**Funding:** This work was supported by the Eunice Kennedy Shriver National Institute of Child Health and Human Development grants: UG1 HD076465 (CLB, AT, AL, MB), UG1 HD078437 (WAC, EC, MM), UG1HD078438 (SJ, SS, RLG), UG1HD076461 (EAL, SLB, OAE, FE, PN), and UG1HD076457 (RJD, MH), and U24HD092094 (TN, JH-F, EMM). Dr. Marion Koso-Thomas is a project officer within the National Institute of Child Health and Human Development (NICHD) Global Network and provided oversight of protocol implementation and monitoring safety and performance. The funders had no role in study design, data collection and analysis, decision to publish, or preparation of the manuscript.

**Competing interests:** The authors have declared that no competing interests exist.

**Abbreviations:** ASPIRIN, Aspirin Supplementation for Pregnancy Indicated risk Reduction In Nulliparas; ITNs, insecticide-treated nets; IPTp, intermittent preventive therapy in pregnancy; GN, Global Network for Womens' and Children's Health Research; DRC, Democratic Republic of the Congo; qPCR, quantitative polymerase chain reaction; BMI, maternal body-mass index; SES, socioeconomic status; PR, prevalence ratio; PD, prevalence difference; DAG, directed acyclic graph; IQR, interquartile range; aPD, adjusted prevalence difference; aPR, adjusted prevalence ratio.

outcomes: preterm birth among Congolese (aPD = 0.06 [99% CI: -0.04, 0.16]), Kenyan (0.03 [-0.04, 0.09]), and Zambian (0.00 [-0.10, 0.20]) women; low birth weight among Congolese (0.07 [-0.03, 0.16]), Kenyan (0.01 [-0.04, 0.06]) and Zambian (-0.04 [-0.13, 0.16]) women; spontaneous abortion among Congolese (0.00 [-0.05, 0.04]), Kenyan (0.00 [-0.04, 0.04]), and Zambian (0.02 [-0.07, 0.24]) women, and anemia later in pregnancy among Congolese (0.04 [-0.09, 0.16]), Kenyan (0.05 [-0.06, 0.17]), and Zambian (0.07 [-0.12, 0.36]) women. The pooled PD for anemia later in pregnancy (26–30 weeks) was 0.08 [99% CI: 0.00, 0.16].

## Conclusions

First-trimester malaria was associated with increased prevalence of anemia later in pregnancy. We identified areas for further investigation including effects of first-trimester malaria on preterm birth and low birth weight.

## Introduction

Malaria in pregnancy is a serious global maternal health problem, with 29% of all pregnancies in sub-Saharan Africa exposed to malaria [1]. Malaria in pregnancy has been associated with preterm birth, low birth weight, and maternal anemia [1, 2]. Because of these adverse effects of malaria in pregnancy, the World Health Organization recommends that all pregnant women in malaria-endemic areas receive effective case management and prevention with the use of insecticide-treated nets (ITNs) [1]. Additionally, in much of sub-Saharan Africa, intermittent preventive therapy (IPTp) with monthly sulfadoxine-pyrimethamine is recommended starting in the second trimester [1]. This strategy of IPTp, though effective in improving birth weight, leaves women and their offspring vulnerable to first-trimester malaria infection.

Most studies assessing the impact of malaria in pregnancy are limited to observations when women begin receiving antenatal care in the second trimester [3]. There are fewer reports of the impact of infections in the first trimester, when placentation may be adversely affected by the known tropism of *Plasmodium falciparum* for placental tissue [3]. In addition to early infection impacting the placentation process, pregnant women have increased susceptibility to malaria, in part due to the parasite's ability to avoid splenic clearance and sequester in the placenta [2, 4]. Our understanding of the biological impact of malaria in early pregnancy is incomplete, and observations in early pregnancy have reported conflicting results on the impact of malaria in early pregnancy on preterm birth, low birth weight, and maternal anemia [5–16]. Most previous studies were single-site, and used wider gestational age intervals to define early pregnancy instead of restricting to the first trimester.

We addressed this gap in knowledge by leveraging a multi-country clinical trial which recruited nulliparous women within the first trimester and followed them throughout pregnancy to obtain maternal and pregnancy outcomes [17, 18]. In the same population, we previously reported the efficacy of low-dose aspirin and the first-trimester malaria prevalence in three sites in sub-Saharan Africa [19, 20]. This report describes an estimate of the effects of first-trimester malaria on maternal and pregnancy outcomes in a sample of women from research sites within three sub-Saharan African countries with varying malaria prevalence. This exploratory analysis uses an efficient trial design (nested within a clinical trial) to examine relationships between malaria in early pregnancy and pregnancy outcomes.

## Materials and methods

### Study design and sample

This sub-study was nested within the *Eunice Kennedy Shriver* NICHD Global Network for Womens' and Children's Health Research (GN)'s trial of low-dose aspirin for the prevention of preterm delivery in nulliparous women with a singleton pregnancy (the ASPIRIN Trial [Clinical Trials Number NCT02409680, first posted April 7, 2015]) [17, 18]. The ASPIRIN trial was a randomized, prospective, multi-national clinical trial that tested the hypothesis that low-dose acetylsalicylic acid, initiated in the first trimester, reduces the risk of preterm birth [18]. The trial recruited 11,976 nulliparous women at seven research sites in six countries (Democratic Republic of the Congo [DRC], Kenya, Zambia, Guatemala, Pakistan, and India) between March 23, 2016 and April 11, 2019 [18]. Women were randomly assigned 1:1 to receive either daily low-dose acetylsalicylic acid (81 mg dose) or a visually identical placebo, beginning in the first trimester (gestational age between 6 weeks, 0 days and 13 weeks, 6 days) and continuing until 36 weeks and 0 days of gestation or delivery [18]. Gestational age was confirmed by study ultrasound. Women were recruited in the first trimester through a plan developed by each site to screen pregnant women who lived within the communities [17].

We recruited the participants of the malaria sub-study from the participants enrolled in the ASPIRIN trial. Women were recruited from both arms of the parent study. Sub-study recruitment occurred as a convenience sample of women from the sub-Saharan African sites (Nord-Ubangi and Sud-Ubangi provinces in the DRC, Bungoma, Busia, and Kakamega counties in Kenya, and Kafue and Chongwe districts in Zambia). Each site had multiple recruitment locations, which include urban and rural settings. Women were enrolled in the malaria sub-study between January 2016 and April 2018. Selection into this convenience sample was based on availability of sample collection supplies, timing of sample transport, and other practical laboratory considerations.

### Measurement of first-trimester malaria

Sample processing procedures have been previously published [20]. Briefly, samples collected at enrollment were tested in duplicate for *P. falciparum* lactate dehydrogenase DNA using quantitative polymerase chain reaction (qPCR) [21]. Samples with discordant results on duplicate testing were excluded from the analysis. The main exposure of interest was first-trimester malaria infection, defined as a positive qPCR result (defined as when florescence for both replicates amplified prior to the 39th cycle or when one replicate did not amplify and the other did prior to the 39th cycle) for *P. falciparum* in a sample obtained during the first trimester. As we recorded first-trimester malaria infection at one time point, we used prevalence of first-trimester malaria infection as our exposure. We are thus including both acute and chronic malaria infection in our exposure variable, and thus our exposure is the *total* impact of PCR prevalence of first-trimester malaria.

### Covariates and outcome assessments

At enrollment, we collected demographics, medical history, and current medical information (including height, weight, blood pressure, heart rate, and history of diabetes) [18]. As a proxy for malnutrition, we used maternal body-mass index (BMI) [22]. Maternal education was categorized as no formal schooling, primary education (1–6 years of schooling), secondary education (7–12 years of schooling), or university and beyond education (≥13 years of schooling). To calculate socioeconomic status (SES), we used the Global Network Socioeconomic Status Index which determines the sum score of 10 specific items owned by the household and

converts this score to a country-specific SES score [23]. We defined the season that coincided with the first trimester of pregnancy as rainy or not rainy, and the rainy season varied across countries: April to October in DRC, April to June and October to November in Kenya, and November to April in Zambia [20].

Maternal and pregnancy outcomes were obtained at delivery and up to 42 days following delivery using the Global Network Maternal and Newborn Health Registry [18]. Preterm birth was defined as a stillbirth or live birth at or after 20 0/7 weeks gestation and before 37 0/7 weeks gestation [18]. Birthweights were measured within 4 days of delivery. Small for gestational age was defined as any live birth whose birth weight was below the INTERGROWTH 10th percentile for a given gestational age and sex of the newborn [18]. Low birth weight was defined as a measured birthweight of < 2500 g [18]. Perinatal mortality was defined as mortality after 20 completed weeks gestation (154 days of gestation) through 7 completed days after birth [18]. Pregnancy losses occurring at 20 weeks gestation or greater were classified as stillbirths and thus included in the perinatal mortality definition [18]. Maternal hemoglobin concentration was measured between 26–30 weeks gestation, and anemia later in pregnancy defined as hemoglobin level < 11 g/dL, based on World Health Organization guidelines for maternal anemia [24].

The primary analysis population included a convenience sample of participants randomized in the ASPIRIN trial who (i) provided any post-baseline outcome data and (ii) delivered at 20 weeks of gestational age or greater [18]. This primary analytic population was consistent with the analytic population used in the ASPIRIN trial. We determined prevalence of preterm birth, small for gestational age, low birth weight, perinatal mortality, and anemia later in pregnancy (26–30 weeks).

We additionally examined the early pregnancy outcome of spontaneous abortion which required a separate analytic population that included women who had pregnancy outcomes before 20 weeks. In this group, we examined spontaneous abortion, defined as premature expulsion of a non-viable fetus from the uterus at less than 20 weeks gestation. For this outcome, we used an analytic population of malaria sub-study participants who had a pregnancy outcome at any time after enrollment [18].

## Statistical analyses

**Crude associations.** To present associations of first-trimester malaria infection and each maternal and pregnancy outcome, we calculated site-specific crude prevalence ratios (PRs) and prevalence differences (PDs) for each maternal and pregnancy outcome by first-trimester malaria infection, using 2x2 tables stratified by exposure and outcome. We present 99% confidence intervals to account for multiple comparisons in order to limit the false positive rate. We did not calculate the crude prevalence ratio for variables in which there were no cases of the outcome in a particular cell. A null value for prevalence difference (i.e., no difference in women with or without first-trimester malaria) is 0 and a null value for prevalence ratio is 1.

Because we anticipated that malaria endemicity impacts maternal and pregnancy outcomes at each site [2], and since first-trimester malaria infection prevalence ranged from 63% in DRC to 6% in Zambia [20], we then determined if we could combine site-specific data to develop summary estimates. Thus, we assessed between-country heterogeneity using the $I^2$ value, and If $I^2$ exceeded a pre-specified threshold of 40% [25], we did not pool results across countries to calculate a summary estimate. If the $I^2$ value was ≤40%, we used the DerSimonian and Laird inverse variance method to calculate a summary estimate [25]. We did not pool results of any outcome that had zero participants in at least one of the cells when stratified by first-trimester malaria infection exposure and country.

**Confounder identification and specification.** We identified possible confounders of first-trimester malaria infection on each maternal and pregnancy outcome by using outcome-specific causal directed acyclic graphs (DAGs) to guide our analyses [26]. We identified the minimally sufficient adjustment set of confounders to estimate the total effect of first-trimester malaria infection on *any* of the assessed maternal and pregnancy outcomes: this set was maternal age, maternal education, SES, malnutrition, season that coincided with the first trimester, and ITNs (**Fig 1**). ITN use was not collected by the ASPIRIN trial, and as we were restricted to data collected in the parent trial, we were unable to adjust for ITN use in our model. ASPIRIN trial arm was randomly allocated without respect to malaria in the first trimester status and thus we did not need to control for the effects of the ASPIRIN study protocol. In addition, anemia in the first trimester was included in our DAG and was not identified in the minimally sufficient set of confounders, thus we did not adjust for anemia in the first trimester in our analyses.

**Adjusted effects.** To estimate effects adjusted for confounders, we used parametric g-computation [27] conducted separately for each outcome under study to estimate marginal (or population average) effects. The mean outcome of malaria in the first trimester on the assessed maternal and birth outcome is the weighted average of the mean outcomes for the combination of values for the confounders included (i.e., the standardized outcome). In order to interpret our results as an estimate of the average causal effect, we assumed counterfactual consistency, exchangeability, and positivity [28]. We first modeled the association of the exposure of first-trimester malaria infection and outcome under study, controlling for confounders, using logistic regression. The logistic regression model produced estimated parameters–the beta coefficients from a regression model. We used those estimated beta coefficients to predict the probability of the outcome for each individual under two scenarios: first, that all

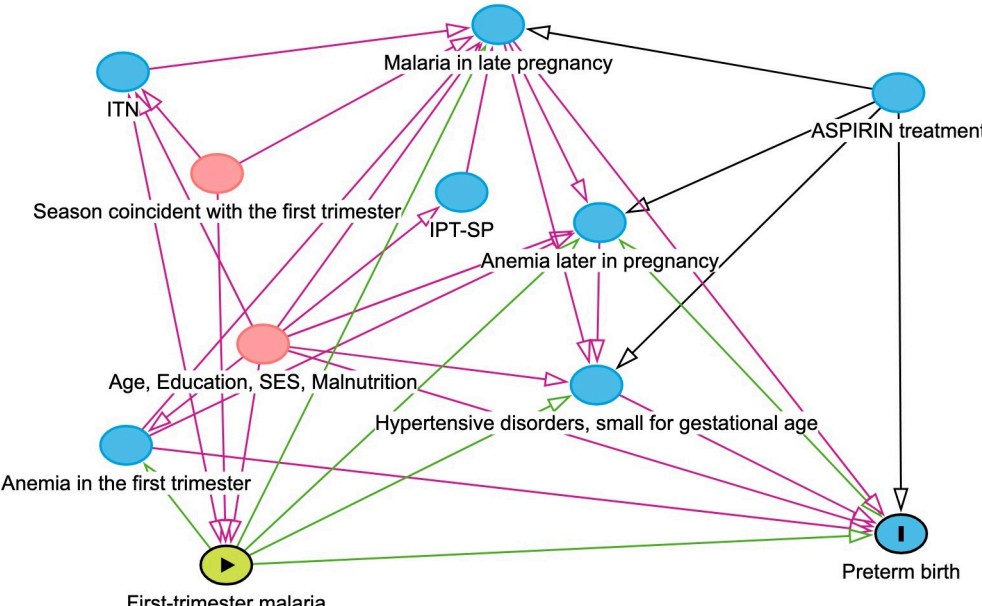

**Fig 1. Directed acyclic graph of the relationship between first-trimester malaria and preterm birth.** The exposure is first-trimester malaria and the outcome is preterm birth. The minimally sufficient adjustment set to determine the total effect of first-trimester malaria on preterm birth is season coincident with the first trimester, maternal age, maternal education, maternal socioeconomic status (SES), malnutrition (used maternal BMI as a proxy), and insecticide-treated nets (ITN) use. Abbreviations: ITN–insecticide treated nets, IPT-SP–intermittent preventative therapy with sulfadoxine-pyrimethamine, SES–socioeconomic status, ASPIRIN–treatment of the ASPIRIN trial (low-dose acetylsalicylic acid or placebo).

participants were exposed to first-trimester malaria infection, and second, that no participants were exposed to first-trimester malaria infection. By doing this, we were able to estimate a mean prevalence of the outcome in two counterfactual exposure situations; we then took a contrast between the mean prevalence if everyone were exposed, and the mean prevalence if everyone were unexposed, to estimate the prevalence difference and prevalence ratio adjusted for confounding (aPD and aPR, respectively). We then used bootstrapping with 10,000 repetitions to determine the corresponding 99 percentile confidence intervals, using the 99% confidence interval to account for multiple comparisons, by selecting the 50th ordered value as the lower limit and 9950th ordered value as the upper limit.

Comparisons were limited to observations without missing data for each variable, and models were limited to observations without missing data for all covariates and the outcome. All analyses were conducted using the R statistical platform (version 4.0.5) [29].

### Ethics approval and consent to participate

The ASPIRIN trial protocol and malaria sub-study protocol were approved by all the sites' and partner institutions' ethics review committees [18]. These committees included the following: Kinshasa School of Public Health, Kinshasa, Democratic Republic of the Congo; University of North Carolina-Chapel Hill, Chapel Hill, North Carolina, United States; University Teaching Hospital, Lusaka, Zambia; University of Alabama at Birmingham, Birmingham, Alabama, United States; Moi University in Eldoret, Kenya; Indiana University School of Medicine, Indianapolis, Indiana, United States; Aga Khan University, Karachi, Pakistan; Columbia University, New York, New York, United States; and RTI International, Durham, North Carolina, United States. Research personnel obtained informed, written consent from all participants [18]. When allowed by the individual ethics review committees of the DRC, Kenya, and Zambia, minors 14 years or younger were enrolled in the ASPIRIN trial [18]. The ASPIRIN trial was first posted April 7, 2015 through Clinical Trials Number NCT02409680.

### Inclusivity in global research

Additional information regarding the ethical, cultural, and scientific considerations specific to inclusivity in global research is included in the Supporting Information (S1 Checklist).

## Results

### Population characteristics

The ASPIRIN trial enrolled 3,800 nulliparous pregnant women from sub-Saharan African sites: 1,362 from DRC, 1,400 from Kenya, and 1,038 from Zambia. From this population, we included a convenience sample of 1,446 women who delivered at 20 weeks gestation or later (469 from DRC, 642 from Kenya, and 335 from Zambia) as our primary analytic population (**Fig 2**). For the outcome of spontaneous abortions, this secondary analytic population was 1,503 women who had a recorded pregnancy outcome since enrollment into the trial.

In the primary analytic population, most participants were under 20 years of age (62%) and were recruited before 12 weeks of gestation (69%) (**Table 1**). The median projected gestational age at enrollment ranged from 10.1 (interquartile range (IQR): 8.3, 12.0) weeks among Kenyan women to 11.4 (IQR: 9.1, 12.7) weeks among Zambian women. Congolese women were younger, shorter, had lower BMI, and had lower educational attainment (fewer with secondary or university education) compared to Kenyan or Zambian women. Most women had a nurse or midwife as a delivery attendant (87%), delivered at a clinic or health center (65%), and had a vaginal delivery (96%).

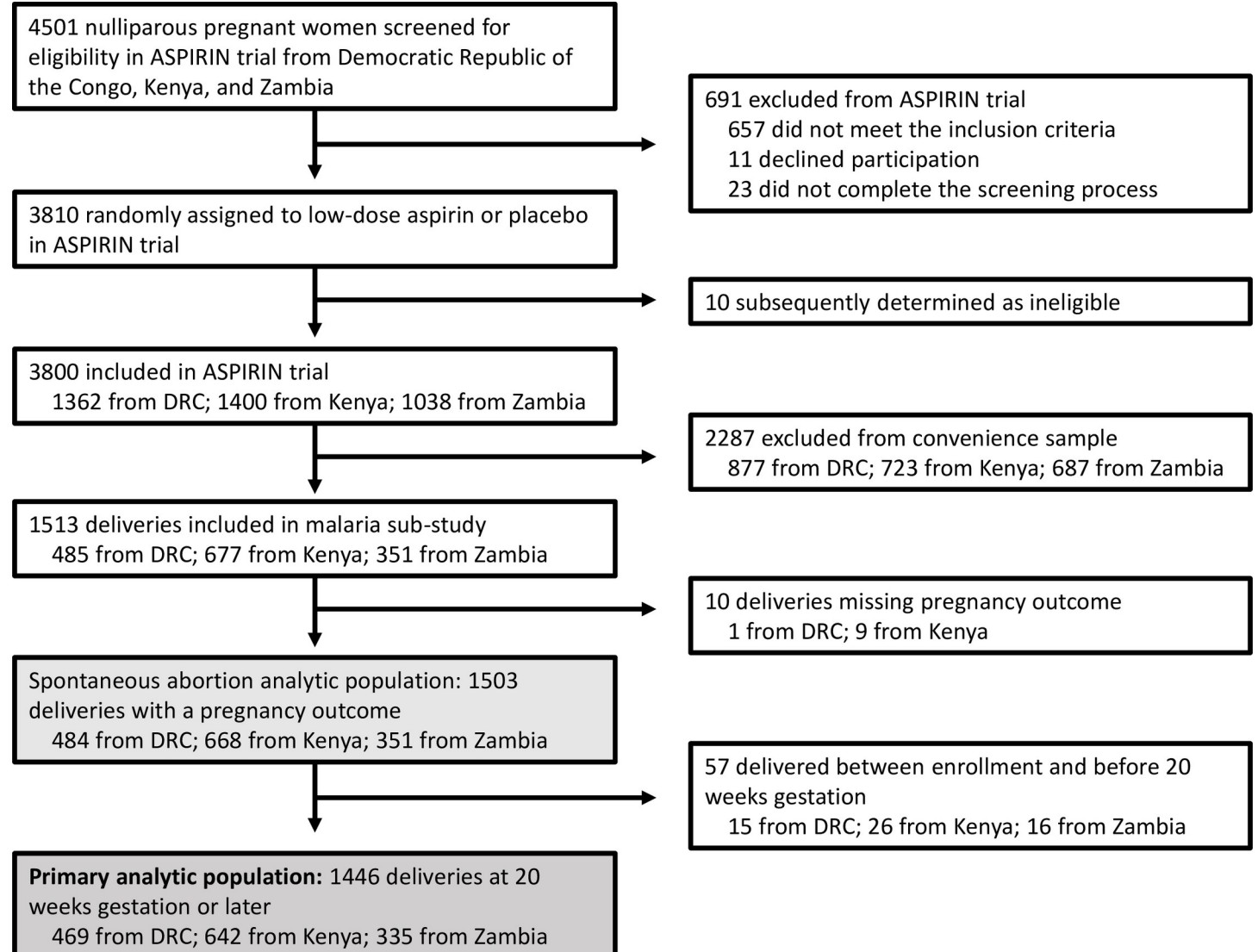

**Fig 2. Study population of malaria analysis sub-study.** The ASPIRIN trial population included 3800 women from Democratic Republic of the Congo (DRC), Kenya, and Zambia. From these women, we took a convenience sample of 1513 deliveries included in the malaria sub-study, and among those, 1446 delivered at 20 weeks gestation or later: 469 from DRC, 642 from Kenya, and 335 from Zambia.

### Prevalence of adverse maternal and pregnancy outcomes

In the primary analytic population, Congolese women had the highest prevalence of preterm birth (19.6%), small for gestational age (15.8%), low birth weight (17.2%), and maternal anemia later in pregnancy (56.8%) (**Table 2**). All three sites had similar prevalence of perinatal mortality.

### Crude associations of first-trimester malaria infection with maternal and pregnancy outcomes

Among the primary analytic population, the first-trimester malaria infection prevalence (as assessed at enrollment) was 63.3% (297/469) among Congolese women, 38.0% (244/642) among Kenyan women, and 6.3% (21/335) among Zambian women (**Table 1**).

**Table 1. Characteristics of study participant population who delivered at 20 weeks gestation or later, by country.**

| Variable | DRC | KENYA | ZAMBIA |
|---|---|---|---|
| **Randomized, N** | 469 | 642 | 335 |
| **Malaria positive, N (%)** | 297 (63.3) | 244 (38.0) | 21 (6.3) |
| **Maternal age (years), N (%)** | | | |
| < 20 | 396 (84.4) | 298 (46.4) | 196 (58.5) |
| 20–29 | 66 (14.1) | 340 (53.0) | 136 (40.6) |
| > 29 | 7 (1.5) | 4 (0.6) | 3 (0.9) |
| Median (P25, P75) | 18.0 (17.0, 18.0) | 20.0 (18.0, 22.0) | 19.0 (18.0. 21.0) |
| **Projected gestation age at enrollment (weeks, days), N (%) [a]** | | | |
| 6, 0–7, 6 | 44 (9.4) | 110 (17.1) | 37 (11.0) |
| 8, 0–9, 6 | 127 (27.1) | 198 (30.8) | 74 (22.1) |
| 10, 0–10, 6 | 68 (14.5) | 86 (13.4) | 33 (9.9) |
| 11, 0–11, 6 | 81 (17.3) | 85 (13.2) | 49 (14.6) |
| 12, 0–13, 6 | 149 (31.8) | 163 (25.4) | 142 (42.4) |
| Median (P25, P75) | 10.9 (9.1, 12.3) | 10.1 (8.3, 12.0) | 11.4 (9.1, 12.7) |
| **Maternal education, N (%)** | | | |
| No formal | 76 (16.2) | 1 (0.2) | 11 (3.3) |
| Primary | 226 (48.2) | 42 (6.5) | 31 (9.3) |
| Secondary | 166 (35.4) | 520 (81.0) | 289 (86.3) |
| University + | 1 (0.2) | 79 (12.3) | 4 (1.2) |
| **Maternal height (cm), mean (SD)** | | | |
| | 155.8 (6.6) | 156.2 (8.8) | 157.5 (6.4) |
| **Maternal weight (kg), mean (SD)** | | | |
| | 50.5 (6.7) | 56.4 (7.4) | 54.6 (8.8) |
| **Maternal BMI (kg/m$^2$), mean (SD)** | | | |
| | 20.8 (2.2) | 23.3 (3.5) | 22.0 (3.3) |
| **Antenatal care visits, mean (SD)** | | | |
| | 3.7 (1.1) | 4.4 (1.3) | 4.0 (0.8) |
| **Delivery attendant, N (%)** | | | |
| Physician | 11 (2.3) | 25 (3.9) | 24 (7.2) |
| Nurse/nurse midwife | 409 (87.2) | 550 (85.7) | 295 (88.1) |
| Traditional birth attendant | 42 (9.0) | 47 (7.3) | 5 (1.5) |
| Family/Self/Other | 7 (1.5) | 20 (3.1) | 11 (3.3) |
| **Delivery location, N (%)** | | | |
| Hospital | 62 (13.2) | 141 (22.0) | 136 (40.6) |
| Clinic/health center | 350 (74.6) | 406 (63.2) | 184 (54.9) |
| Home/Other | 57 (12.2) | 95 (14.8) | 15 (4.5) |
| **Delivery mode, N (%)** | | | |
| Vaginal | 460 (98.1) | 618 (96.3) | 317 (94.6) |
| C-section | 9 (1.9) | 24 (3.7) | 18 (5.4) |

Abbreviations: DRC, Democratic Republic of the Congo; N, number; P25, 25[th] percentile; P75, 75[th] percentile; SD, standard deviation; BMI, body-mass index.

[a] Projected gestational age at enrollment developed from algorithm described in Hoffman et al., 2020.

In the DRC site and at the Kenyan site, the prevalence of preterm birth, small for gestational age, low birth weight, or of perinatal mortality was similar between women with and without first-trimester malaria infection (**Table 3** and **Fig 3**). In the Zambia site, prevalence of preterm

**Table 2. Crude prevalence for maternal and pregnancy outcomes stratified by first-trimester malaria infection status, by country.**

| | DRC | | | KENYA | | | ZAMBIA | | |
|---|---|---|---|---|---|---|---|---|---|
| | Prevalence (%, n/N) | | | Prevalence (%, n/N) | | | Prevalence (%, n/N) | | |
| | Overall | Malaria + | Malaria - | Overall | Malaria + | Malaria - | Overall | Malaria + | Malaria - |
| **Analysis population (pregnancies delivered at 20 weeks of gestation or greater) N = 1446** | | | | | | | | | |
| **Preterm birth** | | | | | | | | | |
| | 19.6 (92/469) | 21.9 (65/297) | 15.7 (27/172) | 9.0 (58/642) | 11.1 (27/244) | 7.8 (31/398) | 6.9 (23/335) | 4.8 (1/21) | 7.0 (22/314) |
| **Small for gestational age** | | | | | | | | | |
| | 15.8 (70/442) | 14.8 (42/283) | 17.6 (28/159) | 8.7 (52/597) | 7.9 (18/227) | 9.2 (34/370) | 14.3 (45/315) | 15.0 (3/20) | 14.2 (42/295) |
| **Low birth weight** | | | | | | | | | |
| | 17.2 (79/458) | 19.9 (58/292) | 12.7 (21/166) | 5.9 (36/609) | 7.4 (17/231) | 5.0 (19/378) | 9.1 (30/330) | 4.8 (1/21) | 9.4 (29/309) |
| **Perinatal mortality** | | | | | | | | | |
| | 6.8 (32/469) | 7.4 (22/297) | 5.8 (10/172) | 5.3 (34/642) | 7.4 (18/244) | 4.0 (16/398) | 5.4 (18/335) | 0.0 (0/21) | 5.7 (18/314) |
| **Anemia later in pregnancy** | | | | | | | | | |
| | 56.8 (256/451) | 58.7 (168/286) | 53.3 (88/165) | 35.5 (185/521) | 41.5 (83/200) | 31.8 (102/321) | 11.8 (33/279) | 23.5 (4/17) | 11.1 (29/262) |
| **Spontaneous abortion (any recorded pregnancy outcome from enrollment to delivery) N = 1503** | | | | | | | | | |
| | 2.7 (13/484) | 2.0 (6/304) | 3.9 (7/180) | 3.7 (25/668) | 3.9 (10/254) | 3.6 (15/414) | 4.6 (16/351) | 4.6 (1/22) | 4.6 (15/329) |

birth, small for gestational age, or low birth weight were similar between women with and without first-trimester malaria infection, and perinatal mortality could not be assessed in the Zambia site due to low numbers.

The prevalence of anemia later in pregnancy was non-significantly higher among women with first-trimester malaria infection compared to women without first-trimester malaria infection in all three site-specific analyses.

The results of the heterogeneity assessment supported pooling across countries for the crude PR and crude PD for preterm birth, small for gestational age, anemia later in pregnancy, and spontaneous abortion, and for the crude PR for low birth weight (Fig 3).

Comparing women from DRC, Kenya, and Zambia with first-trimester malaria infection to those without first-trimester malaria infection, the summary PR for preterm birth was 1.38

**Table 3. Crude associations between first-trimester malaria and maternal and pregnancy outcomes among nulliparous women, by country.**

| | DRC | | KENYA | | ZAMBIA | |
|---|---|---|---|---|---|---|
| | PR (99% CI) | PD (99% CI) | PR (99% CI) | PD (99% CI) | PR (99% CI) | PD (99% CI) |
| **Analysis population (pregnancies delivered at 20 weeks of gestation or greater) N = 1446** | | | | | | |
| **Preterm birth** | | | | | | |
| | 1.39 (0.82, 2.38) | 0.06 (-0.03, 0.16) | 1.42 (0.75, 2.71) | 0.03 (-0.03, 0.10) | 0.68 (0.05, 8.87) | -0.02 (-0.15, 0.10) |
| **Small for gestational age** | | | | | | |
| | 0.84 (0.47, 1.50) | -0.03 (-0.12, 0.07) | 0.86 (0.42, 1.77) | -0.01 (-0.07, 0.05) | 1.05 (0.25, 4.36) | 0.01 (-0.20, 0.22) |
| **Low birth weight** | | | | | | |
| | 1.57 (0.86, 2.88) | 0.07 (-0.02, 0.16) | 1.46 (0.64, 3.37) | 0.02 (-0.03, 0.08) | 0.51 (0.04, 6.53) | -0.05 (-0.17, 0.08) |
| **Perinatal mortality** | | | | | | |
| | 1.27 (0.49, 3.30) | 0.02 (-0.04, 0.08) | 1.84 (0.78, 4.34) | 0.03 (-0.02, 0.08) | NA | -0.06 (-0.09, -0.02) |
| **Anemia later in pregnancy** | | | | | | |
| | 1.10 (0.88, 1.38) | 0.05 (-0.07, 0.18) | 1.31 (0.97, 1.77) | 0.10 (-0.01, 0.21) | 2.13 (0.63, 7.15) | 0.12 (-0.15, 0.39) |
| **Spontaneous Abortion (any recorded pregnancy outcome from enrollment to delivery) N = 1503** | | | | | | |
| | 0.51 (0.12, 2.08) | -0.02 (-0.06, 0.02) | 1.09 (0.39, 3.05) | 0.00 (-0.04, 0.04) | 1.00 (0.07, 13.41) | 0.00 (-0.12, 0.12) |

Abbreviations: PD–crude prevalence difference, PR–crude prevalence ratio, CI–confidence interval, DRC–Democratic Republic of the Congo, NA–not available.

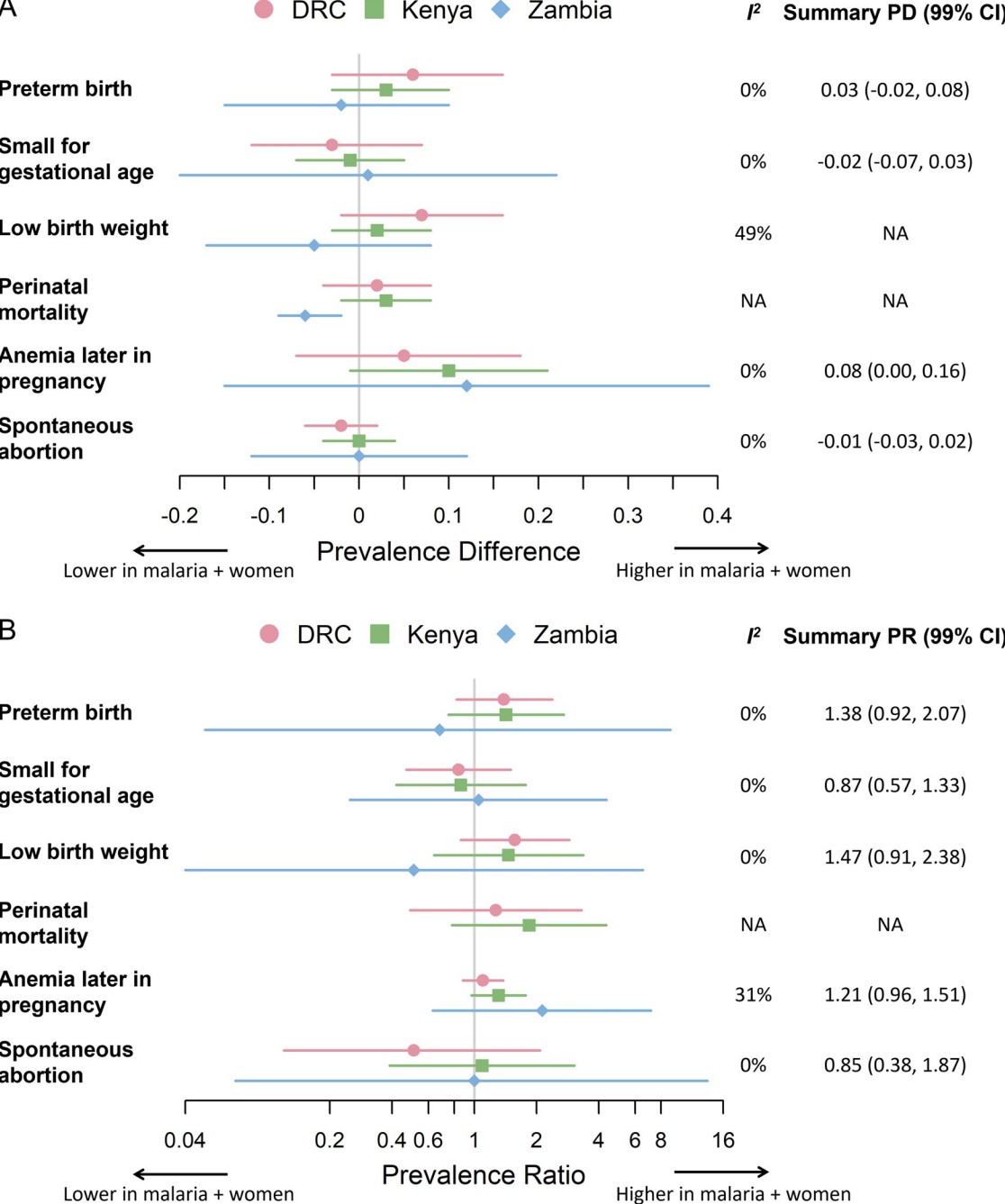

**Fig 3. The impact of first-trimester malaria on adverse maternal and pregnancy outcomes among nulliparous women.**
Comparisons are made using crude prevalence differences (A) or crude prevalence ratios (B), and are stratified by country: DRC (circles), Kenya (squares), and Zambia (diamonds). The error bars are 99% CI. Summary crude estimates were only presented if the $I^2$ value was less than 40% and are presented with 99% CI. The vertical grey line is the null value: 0 for prevalence difference in (A), 1 for prevalence ratio in (B). Abbreviations: CI, confidence interval, DRC, Democratic Republic of the Congo, NA, not available, PD, prevalence difference, PR, prevalence ratio.

[99% CI: 0.92, 2.07], for small for gestational age was 0.87 [0.57, 1.33], for low birth weight was 1.47 [0.91, 2.38], and for spontaneous abortion was 0.85 [0.38, 1.87]. Among Congolese,

Kenyan, and Zambian women, the summary PD of anemia later in pregnancy was significantly higher among those with first-trimester malaria infection compared to those without: 0.08 [0.00, 0.16] (**Fig 3**).

## Adjusted effect of first-trimester malaria infection on maternal and pregnancy outcomes

Comparing women with first-trimester malaria infection to those without, the adjusted prevalence difference (aPD) of preterm birth ranged from 0.06 [99% CI: -0.04, 0.16] among Congolese women to 0.00 [-0.10, 0.20] among Zambian women. The aPD of low birth weight ranged from 0.07 [-0.03, 0.16] among Congolese women to -0.04 [-0.13, 0.16] among Zambian women comparing those with first-trimester malaria infection to those without. The aPD for perinatal mortality was similar between Congolese and Kenyan women (0.02 [-0.04, 0.08] and 0.03 [-0.02, 0.08], respectively) comparing those with and without first-trimester malaria infection, but our ability to determine differences among Zambian women was limited by low perinatal mortality rates among those with first-trimester malaria infection in Zambia. The aPD of anemia later in pregnancy ranged from 0.04 [-0.09, 0.16] among Congolese women to 0.07 [-0.12, 0.36] among Zambian women comparing those with first-trimester malaria infection to those without (**Table 4**).

## Analyses for spontaneous abortion

Participant characteristics were very similar between our primary analytic and secondary analytic populations (**S1 Table**). Among all recorded pregnancy outcomes, the prevalence of spontaneous abortion was highest in the Zambian site (4.6%, 16/351), lower in the Kenyan site (3.7%, 25/668), and lowest in the DRC site (2.7%, 13/484) (**Table 2**). The prevalence of spontaneous abortion was similar among women with and without first-trimester malaria infection in crude stratified site-specific analyses (**Table 3**). In site-specific adjusted analyses, there was

**Table 4. Adjusted associations between first-trimester malaria and maternal and pregnancy outcomes among nulliparous women, by country.**

|  | DRC | | KENYA | | ZAMBIA | |
|---|---|---|---|---|---|---|
|  | aPR (99% CI) | aPD (99% CI) | aPR (99% CI) | aPD (99% CI) | aPR (99% CI) | aPD (99% CI) |
| **Preterm birth** | | | | | | |
| Malaria + vs. Malaria - | 1.49 (0.80, 2.98) | 0.06 (-0.04, 0.16) | 1.36 (0.64, 2.65) | 0.03 (-0.04, 0.09) | 0.99 (0.00, 4.82) | 0.00 (-0.10, 0.20) |
| **Small for gestational age** | | | | | | |
| Malaria + vs. Malaria - | 0.78 (0.41, 1.46) | -0.05 (-0.16, 0.05) | 0.84 (0.36, 1.62) | -0.02 (-0.08, 0.04) | 1.18 (0.00, 3.44) | 0.02 (-0.17, 0.29) |
| **Low birth weight** | | | | | | |
| Malaria + vs. Malaria - | 1.69 (0.86, 3.63) | 0.07 (-0.03, 0.16) | 1.21 (0.48, 2.79) | 0.01 (-0.04, 0.06) | 0.63 (0.00, 3.07) | -0.04 (-0.13, 0.16) |
| **Perinatal mortality** | | | | | | |
| Malaria + vs. Malaria - | 1.63 (0.52, 5.81) | 0.02 (-0.04, 0.08) | 1.80 (0.64, 4.61) | 0.03 (-0.02, 0.08) | NA | -0.06 (-0.09, -0.03) |
| **Anemia later in pregnancy** | | | | | | |
| Malaria + vs. Malaria - | 1.07 (0.86, 1.35) | 0.04 (-0.09, 0.16) | 1.17 (0.83, 1.60) | 0.05 (-0.06, 0.17) | 1.70 (0.00, 5.03) | 0.07 (-0.12, 0.36) |
| **Spontaneous abortion** | | | | | | |
| Malaria + vs. Malaria - | NA | 0.00 (-0.05, 0.04) | 1.13 (0.26, 3.26) | 0.00 (-0.04, 0.04) | 1.64 (0.00, 9.72) | 0.02 (-0.07, 0.24) |

Note: All models were adjusted for age, education, socioeconomic status, season that coincided with the first trimester of pregnancy, and body-mass index.

99% CIs were developed using bootstrapping with 10,000 resamples and selecting the 50th and 9950th value. Many lower limits for the 99% CIs for aPR were 0.00 because at least 50 of the lowest values were 0.00. The CIs for Zambia are not symmetrical because of very few numbers with the exposure assessed.

Abbreviations: aPD–adjusted causal prevalence difference, aPR–adjusted causal prevalence ratio, CI–confidence interval, DRC–Democratic Republic of the Congo, NA–not available.

no statistically significant difference in the prevalence of spontaneous abortion between women who had first-trimester malaria infection and those who did not (**Table 4**).

## Discussion

This is the first study to estimate the effect of first-trimester malaria infection on maternal and pregnancy outcomes among a large number of women from multiple sub-Saharan African sites and transmission settings. Our data suggest an adverse relationship between first-trimester malaria infection and outcomes of preterm birth and low birth weight among Congolese pregnancies. Our data also suggest an adverse relationship between first-trimester malaria infection and the prevalence of anemia later in pregnancy among Congolese, Kenyan, and Zambian women. While these associations were not statistically significant, collectively, the consistent associations we observed suggest that first-trimester malaria infection may result in adverse pregnancy outcomes. Because this study was not designed to be powered to detect significant impact on outcomes, these relationships should be explored in larger populations.

Our study adds to the literature describing malaria and adverse pregnancy outcomes. Previous studies have shown inconsistent associations between malaria infection in early pregnancy and preterm birth, low birth weight, or maternal anemia [5–16]. For example, Moeller et al. found an effect of malaria infection before 15 weeks of gestation on birth weight among 138 Tanzanian women [11], but Accrombessi et al. found no effect of first-trimester malaria infection on birth weight among 273 Beninese women [14].

The possibility that malaria infection in early pregnancy is associated with adverse pregnancy outcomes is of high public health importance. Even if malaria infection in early pregnancy exerts a small effect on adverse pregnancy outcomes, these adverse outcomes are meaningful when evaluated at a population-level given the large number of affected women in the vulnerable population. In 2010, among an estimated 5.6 million first pregnancies that occurred in malaria-endemic areas in Africa, 2.7 million (48.2%) were estimated to have placental infection [30]. Thus, even if the impact of placental infection led to 0.01 increase in prevalence of an outcome, it would lead to about 27,000 additional outcomes among primigravidae, and each prevalence increase of 0.05 would lead to an additional 135,000 outcomes.

Because our study was nested in the ASPIRIN trial, we enrolled only nulliparous women [94% of our study were primigravidae], who have the highest risk of malaria and consequent negative health outcomes. However, as a result, our findings might not be generalizable to pregnant women of all parities. Due to our median enrollment window of 10–11 weeks of gestation, we might have underestimated the effects of malaria in very early pregnancy, a period of gestation during which placentation and organogenesis occur and pregnancy loss is common [2]. In addition, this single sampling point at enrollment may lead to misclassification as potential exposures later in the first trimester would not be classified as malaria infection positive. Through study participation, women enrolled in this study are likely to have attended more antenatal care visits than the general population, and thus have more opportunities to access IPTp, suggesting our results are in the context of standard IPTp coverage. We did not assess setting (rural/urban) in our DAG. In addition, while we assessed malnutrition (using maternal BMI as a proxy), we did not assess the impact of overnutrition leading to maternal overweight which is associated with adverse pregnancy outcomes [31]. We also did not assess collinearity including for SES and maternal education. Finally, parasite detection may have been incomplete, owing to the use of an assay targeting a single copy gene that performs poorly at low densities compared to assays targeting multiple gene copies [32]. Thus, we may not have identified low-density infections that are common in lower transmission sites such as Zambia [33].

## Conclusions

In this multi-site study of first-trimester malaria, we used an efficient study design nested within a clinical trial to estimate the effect of first-trimester malaria on adverse maternal and pregnancy outcomes among nulliparous women across three transmission settings. These first-trimester infections are very common in high transmission settings and might contribute to higher risk of preterm delivery, low birth weight, and anemia later in pregnancy. Future studies on malaria in pregnancy should focus on incident and chronic malaria in the first trimester, associations with symptomatic malaria, and initiation and compliance with IPTp programs. An ongoing intervention trial (ClinicalTrials.gov ID: NCT05757167) will assess how screening and treating malaria infection in the first-trimester impacts adverse pregnancy outcomes. Considering how many pregnancies are exposed to malaria, even small increases in adverse outcomes could lead to many pregnant women and births impacted and thus strategies aimed to reduce the impact of first-trimester malaria might be needed to reduce adverse pregnancy and pregnancy outcomes.

## Supporting information

**S1 Checklist. Inclusivity in global research checklist.**
(DOCX)

**S1 Table. Characteristics of the study participant analysis population for spontaneous abortion, stratified by country.**
(DOCX)

**S1 Fig. Directed acyclic graph of the relationship between first-trimester malaria and small for gestational age.**
(TIF)

**S2 Fig. Directed acyclic graph of the relationship between first-trimester malaria and low birth weight.**
(TIF)

**S3 Fig. Directed acyclic graph of the relationship between first-trimester malaria and perinatal mortality.**
(TIF)

**S4 Fig. Directed acyclic graph of the relationship between first-trimester malaria and anemia later in pregnancy.**
(TIF)

## Acknowledgments

We would like to thank the participants of the ASPIRIN trial.

## Author Contributions

**Conceptualization:** Carl L. Bose, Jennifer Hemingway-Foday, Elizabeth M. McClure, Steven R. Meshnick, Melissa Bauserman.

**Data curation:** Antoinette Tshefu, Adrien Lokangaka, Waldemar A. Carlo, Elwyn Chomba, Musaku Mwenechanya, Edward A. Liechty, Sherri L. Bucher, Osayame A. Ekhaguere, Fabian Esamai, Paul Nyongesa, Saleem Jessani, Sarah Saleem, Robert L. Goldenberg,

Marion Koso-Thomas, Richard J. Derman, Matthew Hoffman, Steven R. Meshnick, Melissa Bauserman.

**Formal analysis:** Sequoia I. Leuba, Daniel Westreich, Andrew F. Olshan, Steve M. Taylor, Janet L. Moore, Tracy L. Nolen, Steven R. Meshnick, Melissa Bauserman.

**Methodology:** Carl L. Bose, Jennifer Hemingway-Foday, Elizabeth M. McClure, Steven R. Meshnick, Melissa Bauserman.

**Supervision:** Daniel Westreich, Andrew F. Olshan, Steve M. Taylor, Steven R. Meshnick, Melissa Bauserman.

**Writing – original draft:** Sequoia I. Leuba, Daniel Westreich, Melissa Bauserman.

**Writing – review & editing:** Sequoia I. Leuba, Daniel Westreich, Carl L. Bose, Andrew F. Olshan, Steve M. Taylor, Antoinette Tshefu, Adrien Lokangaka, Waldemar A. Carlo, Elwyn Chomba, Musaku Mwenechanya, Edward A. Liechty, Sherri L. Bucher, Osayame A. Ekhaguere, Fabian Esamai, Paul Nyongesa, Saleem Jessani, Sarah Saleem, Robert L. Goldenberg, Janet L. Moore, Tracy L. Nolen, Jennifer Hemingway-Foday, Elizabeth M. McClure, Marion Koso-Thomas, Richard J. Derman, Matthew Hoffman, Melissa Bauserman.

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
