## [Decision Letter · Decision Letter 0]

24 May 2024

PONE-D-24-10908Effects on maternal and pregnancy outcomes of first-trimester malaria infection among nulliparous women from Kenya, Zambia, and the Democratic Republic of the CongoPLOS ONE

Dear Dr. Leuba,

Thank you for submitting your manuscript to PLOS ONE. After careful consideration, we feel that it has merit but does not fully meet PLOS ONE’s publication criteria as it currently stands. Therefore, we invite you to submit a revised version of the manuscript that addresses the points raised during the review process.

We look forward to receiving your revised manuscript.

Kind regards,

Raquel Inocencio da Luz, Phd

Academic Editor

PLOS ONE

Reviewers' comments:

Reviewer's Responses to Questions

**Comments to the Author**

1. Is the manuscript technically sound, and do the data support the conclusions?

Reviewer #1: Yes

Reviewer #2: Yes

2. Has the statistical analysis been performed appropriately and rigorously? 

Reviewer #1: Yes

Reviewer #2: Yes

3. Have the authors made all data underlying the findings in their manuscript fully available?

Reviewer #1: Yes

Reviewer #2: No

4. Is the manuscript presented in an intelligible fashion and written in standard English?

Reviewer #1: Yes

Reviewer #2: Yes

5. Review Comments to the Author

Reviewer #1: An interesting and well-written paper. Please address the attached comments - these are areas which require additional clarification for the reader unfamiliar with the parent study and perhaps with the statistical methods used.

Reviewer #2: Review Leuba et al. First Trimester malaria

This is an important and interesting analysis of the (potential) effects of malaria in the first trimester, as a secondary analysis of trial data. Malaria was measured by PCR in the first trimester, and outcomes examined included prematurity, low birthweight, small for gestational age, perinatal mortality, anemia in late pregnancy, and spontaneous abortion. The manuscript is well written although could occasionally be a bit more concise (e.g. line 306-312).

I only have a few comments.

Maternal haemoglobin concentration was measured between 26-30 weeks gestation, if I understand correctly, but is referred to as anemia in late pregnancy. However, 26-30 weeks is closer to anemia in the 2nd trimester. Although this is more about semantics I am wondering if there is a better way to express this so people do not mix this up with anaemia at the end of pregnancy, e.g. say anaemia later in pregnancy or anaemia in 2nd half of pregnancy?

Was setting (urban/peri-urban/rural) considered for inclusion in the daggity diagram? If diagrams were made for the other outcomes as well, perhaps they can be shared in the supplement?

For malaria testing, a sensitive tool was used compared to e.g. a blood smear or rapid diagnostic test. PCR is more likely to pick up asymptomatic malaria, and lower parasite densities. Was there also information on complaints of fever or measured body temperature at enrolment, and have the authors explored if results were the same with a definition of symptomatic malaria? (This type of analysis may only be possible for Kenya and DRC).

The authors used parametric g-computation as statistical tool. It would be nice if the authors can add a line on the advantage of this method compared to a regular (two stage or one stage) meta-analysis.

Note: a dot is missing at the end of a sentence in line 423.

6. PLOS authors have the option to publish the peer review history of their article (what does this mean?). If published, this will include your full peer review and any attached files.

Reviewer #1: **Yes: **Beth A Tippett Barr

Reviewer #2: No

---

## [Author Response · Author response to Decision Letter 0]

7 Aug 2024

Thank you for the opportunity to respond to the academic editor and reviewers. Our responses are in blue (in the document Response to Reviewers).

We would like to thank the reviewer for their request to ensure matching PLOS style requirements and file naming. We have made these changes needed to ensure that we are meeting PLOS ONE style requirements. 

We would like to thank the reviewer their request to complete a copy on inclusivity in global research in our revised manuscript. We have uploaded a completed version of this questionnaire as our Supporting Information in our resubmission of this manuscript. 

We would like to thank the editor for highlighting the requirement that our data will be available on acceptance. We will comply with the data availability procedures of the NICHD Global Network that include data sharing through the N-DASH system, which is publicly available. The data coordinating center (RTI International, Durham, North Carolina, United States) will prepare the data for upload to N-DASH. 

We would like to thank the editor for highlighting this change. We have moved the ethics statement to our Methods section (now Lines 263-276) and deleted it from other sections. 

We would like to thank the editor for their comment on the copyright of Figure 1. We have removed Figure 1 from our manuscript and have updated the following included figure numbers (e.g., Figure 2 is now Figure 1, etc) as needed. 

We would like to thank the editor for their comment to review our reference list. We have reviewed our reference list and confirm it is complete and correct. 

Reviewer #1 Comments: 

Reviewer #1: An interesting and well-written paper. Please address the attached comments - these are areas which require additional clarification for the reader unfamiliar with the parent study and perhaps with the statistical methods used.

Effects on maternal and pregnancy outcomes of first-trimester malaria infection among nulliparous women from Kenya, Zambia, and the Democratic Republic of the Congo.

- Prevalence vs. Incidence in first trimester? Does this make a difference? If you’re including both acute and chronic infection, it may perhaps be masking any effect on pregnancy outcomes

We would like to thank the reviewer for their comment on prevalence versus incidence of first trimester malaria. We are using prevalence as we are recording malaria infection at one time point. We have also highlighted that we are specifically using both acute and chronic malaria infection in our exposure variable by adding to the methods section:

Line 151: As we recorded first-trimester malaria infection at one time point, we used prevalence of first-trimester malaria infection as our exposure. We are thus including both acute and chronic malaria infection in our exposure variable, and thus our exposure is the total impact of PCR prevalence of first-trimester malaria.

- PCR CT threshold doesn’t seem to have been used, just a “positive” – but using a quantitative test, it might yield more precise results if you look at the difference between CT for incident vs latent infection? Or is this supposed to read “qualitative”, and if so, what is the threshold for determining positive?

We used quantitative PCR techniques as described in Doctor et al., Diagn Microbiol Infect Dis 2016, which references more specifics published in Taylor et al., J Clin Microbiol 2010. Taylor et al found that these quantitative PCR techniques had an average cycle threshold (CT) value of 32.63, and could detect 39 parasites per microliter. Following these results, we qualitatively defined samples as ‘positive’ for P. falciparum infection when florescence for both replicates crossed the threshold prior to the 39th cycle, or when one replicate did not amplify and the other crossed the threshold prior to the 39th cycle. We have added more detail (underlined) to our methods section:

Line 148: The main exposure of interest was first-trimester malaria infection, defined as a positive qPCR result (defined as when florescence for both replicates amplified prior to the 39th cycle or when one replicate did not amplify and the other did prior to the 39th cycle) for P. falciparum in a sample obtained during the first trimester.

o In your ‘weaknesses’ paragraph of the discussion section, you talk about the underdetection of malaria as a weakness, but it also could have been a strength if you were detecting only incident infections that would be more likely to be associated with poor pregnancy outcomes…. Thinking ahead to policy implications, do you need to ensure you’re most closely estimating malaria incidence in first trimester and how to address that in a context of late ANC initiation in 2nd trimester? Or are you making an argument that all women of reproductive age would benefit from periodic IPTp, and therefore any malaria infection (acute or chronic) is important?

We thank the reviewer for highlighting these important points. We agree that this is a topic that is ripe for future studies and have expanded the conclusions to include these topics. The new text reads: 

Line 470: Future studies on malaria in pregnancy should focus on incident and chronic malaria in the first trimester, associations with symptomatic malaria, and initiation and compliance with IPTp programs. An ongoing intervention trial (ClinicalTrials.gov ID: NCT05757167) will assess how screening and treating malaria infection in the first-trimester impacts adverse pregnancy outcomes. 

- Tables – please indicate using bold or asterisks the findings which are “significant”

We would like to thank the reviewer for their request to highlight which findings in the tables are ‘significant’. However, as no findings in the tables are significant, we have made no changes. 

- For Plos One, this is the first paper I’ve reviewed that uses PR and PD – for the sake of readers without much Epi training, a brief explanation in the methods of how to interpret the two may be helpful, including how to interpret CI’s – much as one does for OR’s, where a CI crossing ‘1’ means the findings are not statistically different in exposed than unexposed.

We would like to thank the reviewer for highlighting the need on how to briefly interpret PR and PD. In addition to highlighting the null value in the caption of Figure 3 (formerly Figure 4), we have also added to our methods section: 

Line 204: A null value for prevalence difference (i.e., no difference in women with or without first-trimester malaria) is 0 and a null value for prevalence ratio is 1.

- Anemia in late pregnancy – did your regression analysis control for anemia in early pregnancy? The implication in the narrative is that you’re looking for anemia that developed during pregnancy (an outcome) vs women who were already anemic at enrolment. This seems like a variable you would want to understand a little more that is currently shown

We would like to thank the reviewer for highlighting that our regression analysis did not control for anemia in early pregnancy. We have clarified this in our methods section:

Line 227: In addition, anemia in the first trimester was included in our DAG and was not identified in the minimally sufficient set of confounders, thus we did not adjust for anemia in the first trimester in our analyses.

- Your methods aren’t clear about whether the women came from both arms of the parent study, and if so, did you control for aspirin exposure in your analysis? Why or why not?

We would like to thank the reviewer for their question on clarifying whether women came from both arms of the parent study and if we controlled for aspirin exposure in our analysis. We thus have added:

Line 134: Women were recruited from both arms of the parent study.

Line 226: ASPIRIN trial arm was randomly allocated without respect to malaria in the first trimester status and thus we did not need to control for the effects of the ASPIRIN study protocol. 

- Did you assess multicollinearity as well as confounders? Specifically, SES and maternal education?

We would like to thank the reviewer for highlighting our limitation that we did not assess multicollinearity specifically for SES and maternal education. We thus have added to our limitations in the discussion. 

Line 458: We also did not assess collinearity including for SES and maternal education.

- Line 231 – what proxy of maternal BMI was used for malnutrition? Be specific. And did you look at overnutrition as well? What was the rationale for not separating nutritional status into three categories? Overnutrition is becoming far more prevalent across Africa, including in rural areas, and has been associated more strongly with poor pregnancy outcomes than undernutrition

We would like to thank the reviewer for their comment on what proxy of maternal BMI was used for malnutrition. This was a typo, and now has been corrected to: 

Line 235: ‘malnutrition (used maternal BMI as a proxy)’

We would like to thank the reviewer for pointing out the possibility of overnutrition impacting our results. Within these research sites, the incidence of overnutrition (as measured by BMIs > 25) is very low (other studies report <5% in the DRC [Bauserman et al., Am J Clin Nutr 2021]). Therefore, we are limited in our ability to make conclusions about the impact of overnutrition in any of our models, and restricted these analyses to evaluation of BMI as a continuous variable. We did not look at overnutrition and agree with the reviewer that this is a limitation of our work. We thus have added to our limitations in the discussion:

Line 455: In addition, while we assessed malnutrition (using maternal 

---

## [Decision Letter · Decision Letter 1]

26 Aug 2024

Effects on maternal and pregnancy outcomes of first-trimester malaria infection among nulliparous women from Kenya, Zambia, and the Democratic Republic of the Congo

PONE-D-24-10908R1

Dear Dr. Leuba,

We’re pleased to inform you that your manuscript has been judged scientifically suitable for publication and will be formally accepted for publication once it meets all outstanding technical requirements.

Kind regards,

Ochuwa Adiketu Babah, FWACS, FMCOG

Academic Editor

PLOS ONE

Additional Editor Comments (optional):

Reviewers' comments:

Reviewer's Responses to Questions

**Comments to the Author**

1. If the authors have adequately addressed your comments raised in a previous round of review and you feel that this manuscript is now acceptable for publication, you may indicate that here to bypass the “Comments to the Author” section, enter your conflict of interest statement in the “Confidential to Editor” section, and submit your "Accept" recommendation.

Reviewer #2: All comments have been addressed

2. Is the manuscript technically sound, and do the data support the conclusions?

Reviewer #2: Yes

3. Has the statistical analysis been performed appropriately and rigorously? 

Reviewer #2: Yes

4. Have the authors made all data underlying the findings in their manuscript fully available?

Reviewer #2: Yes

5. Is the manuscript presented in an intelligible fashion and written in standard English?

Reviewer #2: Yes

6. Review Comments to the Author

Reviewer #2: The authors have addressed the comments satisfactorily. It took me ages to realize that they describe a two-stage meta-analysis, and that is appropriate for the study.

7. PLOS authors have the option to publish the peer review history of their article (what does this mean?). If published, this will include your full peer review and any attached files.

Reviewer #2: No

---

## [Editor Report · Acceptance letter]

10 Oct 2024

PONE-D-24-10908R1 

PLOS ONE

Dear Dr. Leuba, 

I'm pleased to inform you that your manuscript has been deemed suitable for publication in PLOS ONE. Congratulations! Your manuscript is now being handed over to our production team.

Kind regards, 

on behalf of

Dr. Ochuwa Adiketu Babah 

Academic Editor

PLOS ONE